*Report*

**EMBO** *reports*

# CUL2^{LRR1}, TRAIP and p97 control CMG helicase disassembly in the mammalian cell cycle

Fabrizio Villa[1,†] (ID), Ryo Fujisawa[1,†] (ID), Johanna Ainsworth[1], Kohei Nishimura[1,2], Michael Lie-A-Ling[3] (ID), Georges Lacaud[3] (ID) & Karim PM Labib[1,*] (ID)

## Abstract

The eukaryotic replisome is disassembled in each cell cycle, dependent upon ubiquitylation of the CMG helicase. Studies of *Saccharomyces cerevisiae*, *Caenorhabditis elegans* and *Xenopus laevis* have revealed surprising evolutionary diversity in the ubiquitin ligases that control CMG ubiquitylation, but regulated disassembly of the mammalian replisome has yet to be explored. Here, we describe a model system for studying the ubiquitylation and chromatin extraction of the mammalian CMG replisome, based on mouse embryonic stem cells. We show that the ubiquitin ligase CUL2^{LRR1} is required for ubiquitylation of the CMG-MCM7 subunit during S-phase, leading to disassembly by the p97 ATPase. Moreover, a second pathway of CMG disassembly is activated during mitosis, dependent upon the TRAIP ubiquitin ligase that is mutated in primordial dwarfism and mis-regulated in various cancers. These findings indicate that replisome disassembly in diverse metazoa is regulated by a conserved pair of ubiquitin ligases, distinct from those present in other eukaryotes.

**Keywords** DNA replication; CMG helicase; CUL2^{LRR1}; TRAIP; p97 ATPase

**Subject Categories** DNA Replication, Recombination & Repair; Post-translational Modifications & Proteolysis

## Introduction

The eukaryotic replisome is assembled and disassembled in a highly regulated fashion during each cell cycle, to ensure that each chromosome is duplicated just once (Bell & Labib, 2016; Burgers & Kunkel, 2017; Gasser, 2019). The central feature of replisome construction and dissolution is the regulated assembly and disassembly of the CMG DNA helicase (CMG = CDC45_MCM2-7_GINS), around which other factors are recruited to form the replisome at DNA replication forks (Gambus *et al*, 2006; Moyer *et al*, 2006; Gambus *et al*, 2009; Sengupta *et al*, 2013; Pellegrini & Costa, 2016;

Bai *et al*, 2017). The CMG helicase is assembled in a two-step process during G1-phase and S-phase, and then is disassembled during DNA replication termination. CMG disassembly occurs whenever two replication forks meet, or when one fork reaches a DNA end and CMG is released from its DNA template (Deegan *et al*, 2020; Dewar *et al*, 2015; Maric *et al*, 2014; Moreno *et al*, 2014; Vrtis *et al*, 2021).

The catalytic core of CMG is a hetero-hexameric ring of the six MCM2-7 ATPases. At origins of DNA replication during G1-phase, two MCM2-7 rings are loaded in a concerted fashion around double-strand DNA (dsDNA) to form an inactive MCM2-7 double hexamer (Evrin *et al*, 2009; Remus *et al*, 2009; Gambus *et al*, 2011). Subsequently, CDC45 and GINS are recruited to the MCM2-7 double hexamer during the initiation of DNA replication, producing two nascent CMG helicases around dsDNA (Gambus *et al*, 2006). The nascent CMG helicases are then activated, by extrusion of one of the two parental DNA strands from the MCM2-7 ring (Douglas *et al*, 2018), so that CMG encircles the single-strand DNA (ssDNA) template for leading strand synthesis at each replication fork (Fu *et al*, 2011).

The CMG helicase is a metastable complex that is topologically trapped around parental ssDNA throughout the elongation phase of DNA replication (Lewis *et al*, 2020). Chromatin is disrupted ahead of the CMG helicase and non-nucleosomal proteins are displaced, without breaking the association of CMG with replication fork DNA (Kurat *et al*, 2017; Sparks *et al*, 2019). Nevertheless, the encounter of two DNA replication forks leads very rapidly to CMG disassembly and thus to dissolution of the two replisomes, which represents the final stage of eukaryotic chromosome replication (Maric *et al*, 2014; Moreno *et al*, 2014; Dewar *et al*, 2015).

During DNA replication termination, the key regulated step that induces CMG disassembly is ubiquitylation of the MCM7 subunit of the helicase (Maric *et al*, 2014; Moreno *et al*, 2014), which drives recruitment of the Cdc48 / p97 ATPase (Maric *et al*, 2017; Deegan *et al*, 2020). Work with budding yeast indicates that Cdc48 / p97 unfolds ubiquitylated MCM7 and thereby disrupts CMG in an irreversible fashion into CDC45, GINS and subcomplexes of MCM2-7 (Mukherjee & Labib, 2019; Deegan *et al*, 2020).

1  The MRC Protein Phosphorylation and Ubiquitylation Unit, School of Life Sciences, University of Dundee, Dundee, UK
2  Division of Biological Science, Graduate School of Science, Nagoya University, Nagoya, Japan
3  Cancer Research U.K. Manchester Institute, The University of Manchester, Alderley Park, UK
   *Corresponding author. Tel: +44 1382 384108; E-mail: kpmlabib@dundee.ac.uk
   †These authors contributed equally to this work

The ubiquitin ligases that induce CMG ubiquitylation are key regulators of CMG disassembly. The first such ligase was identified in budding yeast, in which the cullin ligase SCF$^{Dia2}$ drives CMG ubiquitylation during DNA replication termination (Maric *et al*, 2014). However, the F-box protein Dia2, which represents the substrate adaptor of SCF$^{Dia2}$, is only found in yeasts. This indicated that the regulated disassembly of the CMG helicase has diversified during eukaryotic evolution, to a greater degree than the initiation process that involves a conserved set of factors (Hills & Diffley, 2014; Zegerman, 2015). In subsequent work, an unrelated cullin ligase called CUL2$^{LRR1}$ (Merlet *et al*, 2010; Burger *et al*, 2013) was found to be required for the ubiquitylation of CMG-MCM7 during DNA replication termination in the nematode *Caenorhabditis elegans* (Sonneville *et al*, 2017) and the frog *Xenopus laevis* (Dewar *et al*, 2017; Sonneville *et al*, 2017).

Surprisingly, studies of *C. elegans* CUL-2$^{LRR-1}$ also revealed the existence of a second pathway for CMG disassembly that had not previously been observed in budding yeast (Sonneville *et al*, 2017). Upon depletion of *C.e.*LRR-1, CMG remained on chromatin after DNA replication termination, but was then extracted and disassembled upon entry into mitosis. A similar reaction was observed when *Xenopus* egg extracts lacking CUL2$^{LRR1}$ activity were driven into mitosis by premature activation of Cyclin-Dependent Kinase or CDK (Deng *et al*, 2019; Priego Moreno *et al*, 2019). In both worm and frog, mitotic CMG disassembly requires a metazoan-specific RING ubiquitin ligase known as TRUL-1 in *C. elegans* (TRUL Ubiquitin Ligase 1) and TRAIP in vertebrates (Deng *et al*, 2019; Priego Moreno *et al*, 2019; Sonneville *et al*, 2019). Importantly, the TRUL-1 / TRAIP pathway of mitotic CMG disassembly in *C. elegans* and *X. laevis* is activated by mitosis but does not require DNA replication termination. Thus, CMG disassembly still occurs if *C. elegans* cells enter mitosis before the completion of DNA replication (Sonneville *et al*, 2019), and the same is true if *Xenopus* egg extracts are induced to enter mitosis with incompletely replicated DNA (Deng *et al*, 2019; Priego Moreno *et al*, 2019). Loss of TRUL-1 in *C. elegans* causes reduced viability in combination with a mutation impairing DNA replication (Sonneville *et al*, 2019), suggesting that the mitotic CMG disassembly pathway evolved to process any remaining sites of incomplete DNA replication before anaphase, in order to facilitate the preservation of genome integrity in dividing cells.

Until now, the factors that mediate CMG ubiquitylation and disassembly in the mammalian cell cycle had not been explored. Human CUL2$^{LRR1}$ was identified as a negative regulator of signalling from a tumour necrosis factor receptor (Jang *et al*, 2001; Kamura *et al*, 2004) and also promotes degradation of a cytoplasmic pool of the CDK inhibitor p21 (Starostina *et al*, 2010). However, nuclear functions for mammalian CUL2$^{LRR1}$ have not previously been reported. In contrast, TRAIP is a nuclear protein that is known to be an important guardian of genome integrity in mammalian cells (Chapard *et al*, 2014; Harley *et al*, 2016; Hoffmann *et al*, 2016; Soo Lee *et al*, 2016; Han *et al*, 2019; Li *et al*, 2020). TRAIP is recruited to sites of DNA damage by a PCNA-Interacting-Peptide (PIP box) at its carboxyl terminus and is required for effective DNA repair and DNA damage signalling, interacting with several repair factors or mediators of the DNA damage response (Wallace *et al*, 2014; Hoffmann *et al*, 2016; Soo Lee *et al*, 2016; Han *et al*, 2019). TRAIP is essential for mouse development (Park *et al*, 2007), whereas RING mutations or truncating mutations in human TRAIP are viable but lead to a

form of primordial dwarfism that is associated with a defective DNA damage response (Harley *et al*, 2016).

A role for human TRAIP in the processing of stalled DNA replication forks was indicated by defects in fork recovery, after the induction of specific forms of DNA damage in cells lacking TRAIP activity (Harley *et al*, 2016; Hoffmann *et al*, 2016). Moreover, recent work indicates that TRAIP is required for the mitotic processing of sites of incomplete DNA replication in human cells, in order to preserve genome integrity (Sonneville *et al*, 2019). However, mammalian TRAIP has yet to be shown to regulate CMG helicase disassembly either during S-phase or mitosis.

Similarly, the role of the p97 ATPase in replisome disassembly has yet to be addressed in mammalian cells. One important reason for the lack of progress with studies of replisome disassembly in mammalian cells has been the lack of a tractable model system. Here, we describe such a system based on mouse embryonic stem cells (mouse ES cells), revealing a conserved role in CMG disassembly for CUL2$^{LRR1}$, TRAIP and p97.

# Results and Discussion

## Mouse ES cells represent an ideal model system with which to study the regulation of CMG disassembly in the mammalian cell cycle

Mouse ES cells such as E14TG2a are easily grown in the absence of feeder cells (see Materials and Methods), have a stable diploid karyotype and support efficient genome editing. Compared with diploid human cell lines such as RPE1, mouse ES cells have a number of features that facilitate studies of the CMG helicase at mammalian DNA replication forks. The majority of asynchronously grown mouse ES cells contain replisomes, since around 70% of cells are in S-phase in serum-containing medium (Appendix Fig S1A and B, Ter Huurne *et al*, 2017). By comparison, only around 35% of asynchronous human RPE cells are present in S-phase (Appendix Fig S1A and B, Matson *et al*, 2017). In addition, it is easy to grow large numbers of mouse cells in order to purify the CMG replisome, since E14TG2a cells have a doubling time of around 12.5 h (see below, Fig EV4F). Moreover, the cells are relatively small and lack contact inhibition (Burdon *et al*, 2002), so that relatively few culture dishes are required to grow a given number of cells. In contrast, human RPE cells have a doubling time of over 20 h (Matson *et al*, 2017), are considerably larger than mouse ES cells and are subject to contact inhibition. Finally, we note that large areas of constitutive heterochromatin in mouse ES cells (Saksouk *et al*, 2015) provide an ideal system with which to monitor replisome components on chromatin in live cells, as discussed below.

We used CRISPR-Cas9 to modify both alleles of the endogenous locus encoding the SLD5 subunit of GINS in E14TG2a cells, in order to introduce a Tandem Affinity Purification (TAP) tag or Green Fluorescent Protein (GFP) at the amino terminus of the SLD5 protein (Figs 1A–F and EV1). Both GFP-SLD5 and TAP-SLD5 co-purified with CDC45, the six MCM2-7 proteins and other replisome subunits (Figs 1G–J, 2A, and EV1). Moreover, whilst tagged SLD5 co-purified with other GINS subunits throughout the cell cycle, the association of GINS with other replisome factors such as MCM2-7 was restricted to S-phase and was only detected upon release from DNA (Fig 1H–J).

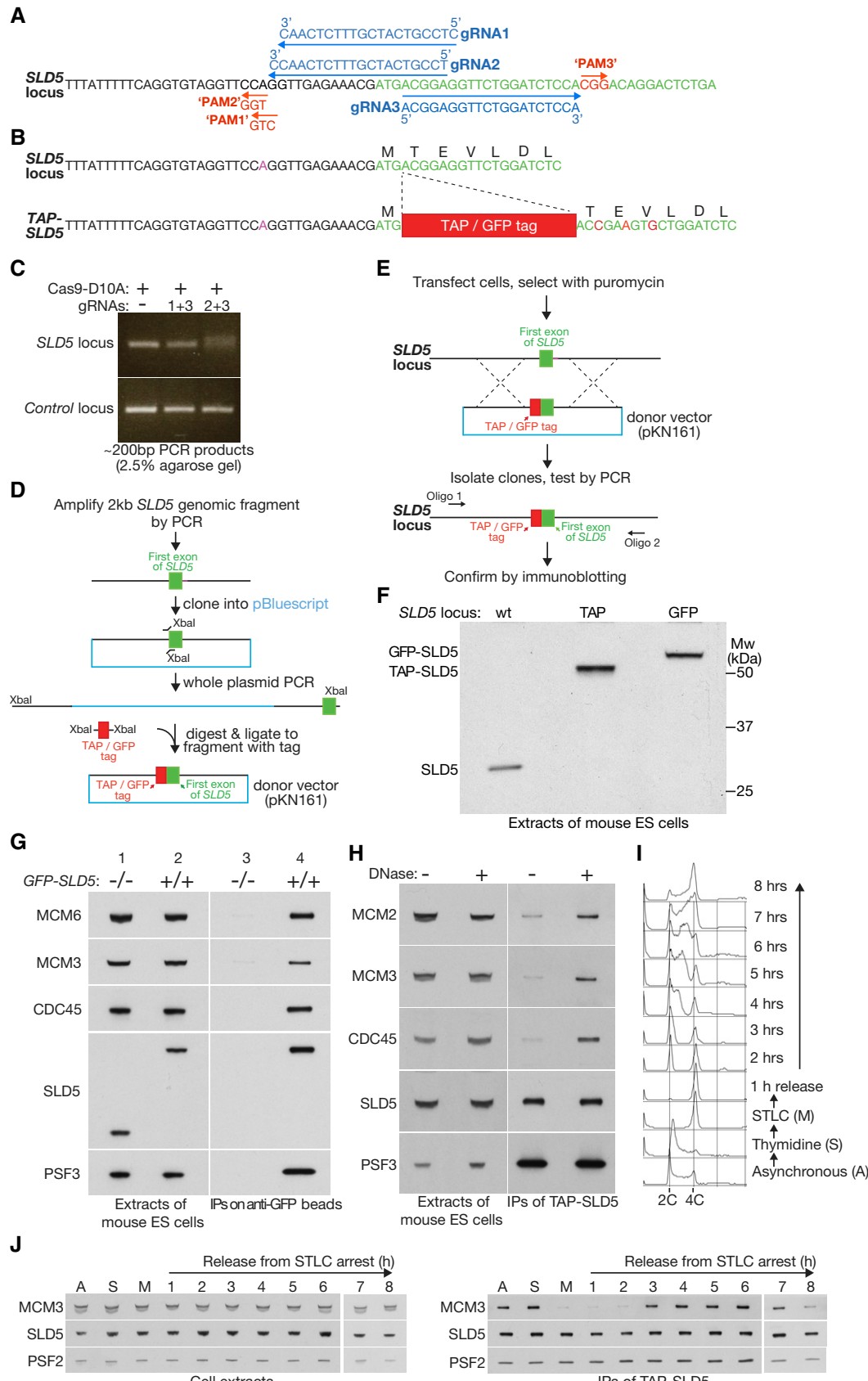

**Figure 1.**

**Figure 1. Mouse ES cells provide a model system for studying the mammalian CMG helicase.**

A   Guide RNAs used to target the 5' end of exon 1 of the *SLD5* gene in mouse ES cells. Each of the targeted sites contains 20 nt homology to the corresponding gRNA, followed by a 3 nt "Protospacer Adjacent Motif" of PAM that has the form "NGG" and is also required for cleavage by Cas9. Note that the predicted PAM site of gRNA1 does not match the "NGG" consensus, due to a polymorphism in E14TG2a ES cells, in comparison to the reference mouse genome sequence. This polymorphism prevents cleavage by Cas9 in combination with gRNA1 (see below).

B   The TAP or GFP tag was inserted after the initiator methionine of SLD5. The donor DNA also contained silent mutations in the first few residues of SLD5 (indicated in red), to prevent the modified locus being cut by Cas9.

C   PCR analysis of a pool of transfected cells after treatment with Cas9-D10A "nickase" and the indicated combination of gRNAs from (A). The ~ 200 bp product was centred around the start of exon 1 of the *SLD5* gene. Efficient cutting of both template strands by Cas9 nickase was indicated by the production of a smeary PCR product (gRNAs 2 + 3), reflecting DNA repair that mostly produced small deletions around the nicking site (e.g. see Fig EV4D below). The combination of gRNAs 1 + 3 did not produce a smeary PCR, due to a polymorphism within the PAM motif of the genomic locus (see A above).

D   Outline of the construction of donor vectors with TAP or GFP tags flanked by 1 kb of 5' and 3' homology sequences.

E   Strategy for generation of mouse ES clones with tagged SLD5 (TAP or GFP).

F   Immunoblot indicating successful tagging of both alleles of *SLD5* with TAP or GFP.

G   Asynchronous cultures of the indicated cells were used to generate cell extracts, which were then incubated with agarose beads coupled to anti-GFP nanobodies. The indicated factors were monitored by immunoblotting.

H   Extracts of *TAP-SLD5* mouse ES cells were incubated plus or minus DNase for 30 min at 4°C, before immunoprecipitation of TAP-SLD5.

I   *TAP-SLD5* cells were synchronised as indicated (further details in Materials and Methods). DNA content was monitored throughout the experiment by flow cytometry.

J   Samples from (C) were taken at the indicated times and used to isolate TAP-SLD5 by immunoprecipitation on IgG beads. The indicated proteins were monitored by immunoblotting. For reasons of space, samples A, S, M and 1–6 were resolved in a separate gel to samples 7–8.

These findings illustrate that the tagged SLD5 subunit of GINS in mouse ES cells provides a useful tool with which to isolate the mammalian CMG helicase and associated replisome factors from DNA replication forks.

## p97 activity is required for chromatin extraction and disassembly of the CMG replisome in mammalian cells

When cells were treated with the p97 inhibitor CB-5083 (Anderson *et al*, 2015), the CMG helicase accumulated and a ladder of ubiquitylated MCM7 was detected (Fig 2A, lane 6; Fig EV2A; ubiquitylation of other CMG subunits was not observed). These findings are consistent with previous observations in yeast (Maric *et al*, 2014), worm (Sonneville *et al*, 2017) and frog (Moreno *et al*, 2014), and indicated that ubiquitylated CMG was stabilised upon inactivation of p97.

To monitor the chromatin association of the CMG replisome in individual cells throughout the cell cycle, we took advantage of the fact that mouse cells have large sub-nuclear bodies of constitutive heterochromatin (Saksouk *et al*, 2015), which represent pericentromeric regions of repetitive DNA sequences and are associated with factors such as HP1 (Fig 2B). The condensed nature of such loci suggested that they might facilitate the visualisation of CMG and other replisome components on interphase chromatin, during the narrow window of time in late S-phase when such heterochromatic regions are replicated.

To explore this idea, we monitored GFP-SLD5 by spinning disk confocal microscopy, in mouse ES cells that also expressed mCherry-tagged PCNA, which is a marker of active sites of DNA synthesis (Shibahara & Stillman, 1999; Leonhardt *et al*, 2000; Essers *et al*, 2005). Consistent with previous reports (Bravo & Macdonald-Bravo, 1987; Leonhardt *et al*, 2000), mCherry-PCNA was observed in many small sub-nuclear foci during early S-phase (Figs EV3A early S, and 2C). Subsequently, mCherry-PCNA relocated during late S-phase to larger patches (Figs EV3A late S, and 2C), which corresponded to constitutive heterochromatin (Fig EV3B). GFP-SLD5 was hard to detect at the smaller PCNA foci in early S-phase, probably reflecting the fact that each fork is associated with just one CMG helicase but multiple PCNA complexes, with the latter remaining on nascent chromatin for a short period after passage of the fork. However, GFP-SLD5 was readily detected on heterochromatic patches during late S-phase, appearing and disappearing with similar kinetics to mCherry-PCNA (Fig 2C). These findings likely reflect the clustering within each heterochromatic patch of many DNA replication forks and indicate that constitutive heterochromatin provides a system with which to monitor the assembly and disassembly of the CMG helicase in mouse ES cells.

**Figure 2. The p97 ATPase is required for chromatin extraction and disassembly of ubiquitylated CMG helicase in mouse ES cells.**

A   Cells were treated with 5 μM CB-5083 as indicated (p97i = p97 inhibitor). Extracts were then incubated with IgG beads to isolate the GINS complex via TAP-SLD5 and the indicated factors were monitored by immunoblotting. Accumulation of the CMG helicase upon inhibition of p97 was reflected by the increased association of GINS with CDC45 and MCM2-7.

B   Immunostaining of HP1 protein in fixed mouse ES cells that were stained with Hoechst to reveal condensed patches of heterochromatic DNA (examples marked by arrows; co-localisation of HP1 and condensed DNA in 83% of cells, *n* = 103).

C   Time-lapse analysis of mouse ES cells expressing GFP-SLD5 and mCherry-PCNA (random integration of mCherry-PCNA as described in Materials and Methods). Arrows indicate co-localisation on heterochromatic patches, which was observed in 14% of asynchronous cells (*n* = 238). GFP-SLD5 appeared and disappeared with similar kinetics to mCherry-PCNA (100% cells, *n* = 101).

D   GFP-SLD5 mouse ES cells were treated with p97i as indicated and then fixed and stained with Hoechst. Arrows indicate accumulation of GFP-SLD5 on constitutive heterochromatin (53% cells, *n* = 125).

E   Immunofluorescence of the replisome components CLASPIN (i) and CTF4 (ii) in GFP-SLD5 cells treated with p97i. Arrows indicate co-localisation of GFP-SLD5 patches with CLASPIN (95% cells, *n* = 107) or CTF4 (91% cells, *n* = 45). Scale bars in all panels correspond to 10 μm.

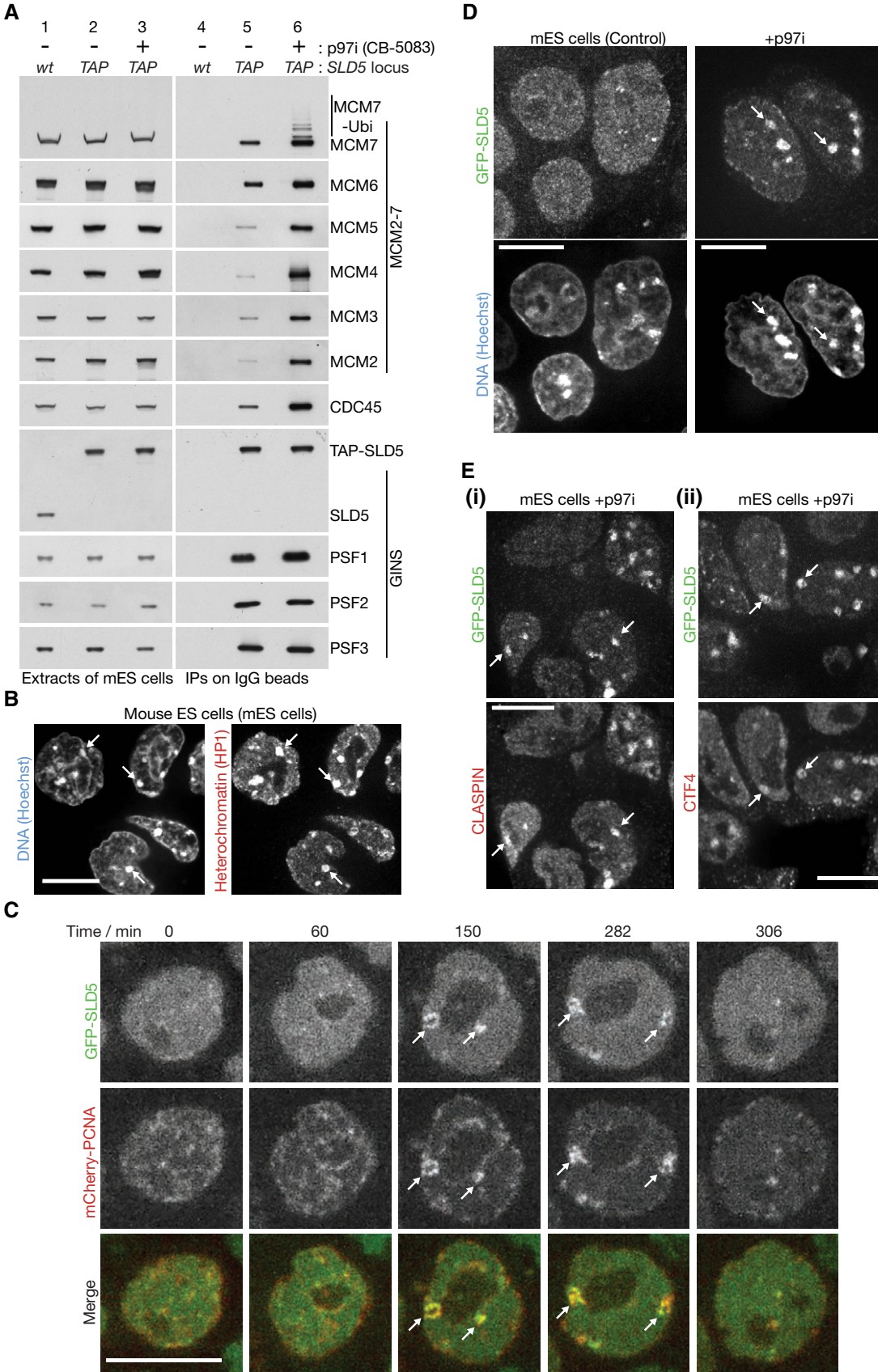

**Figure 2.**

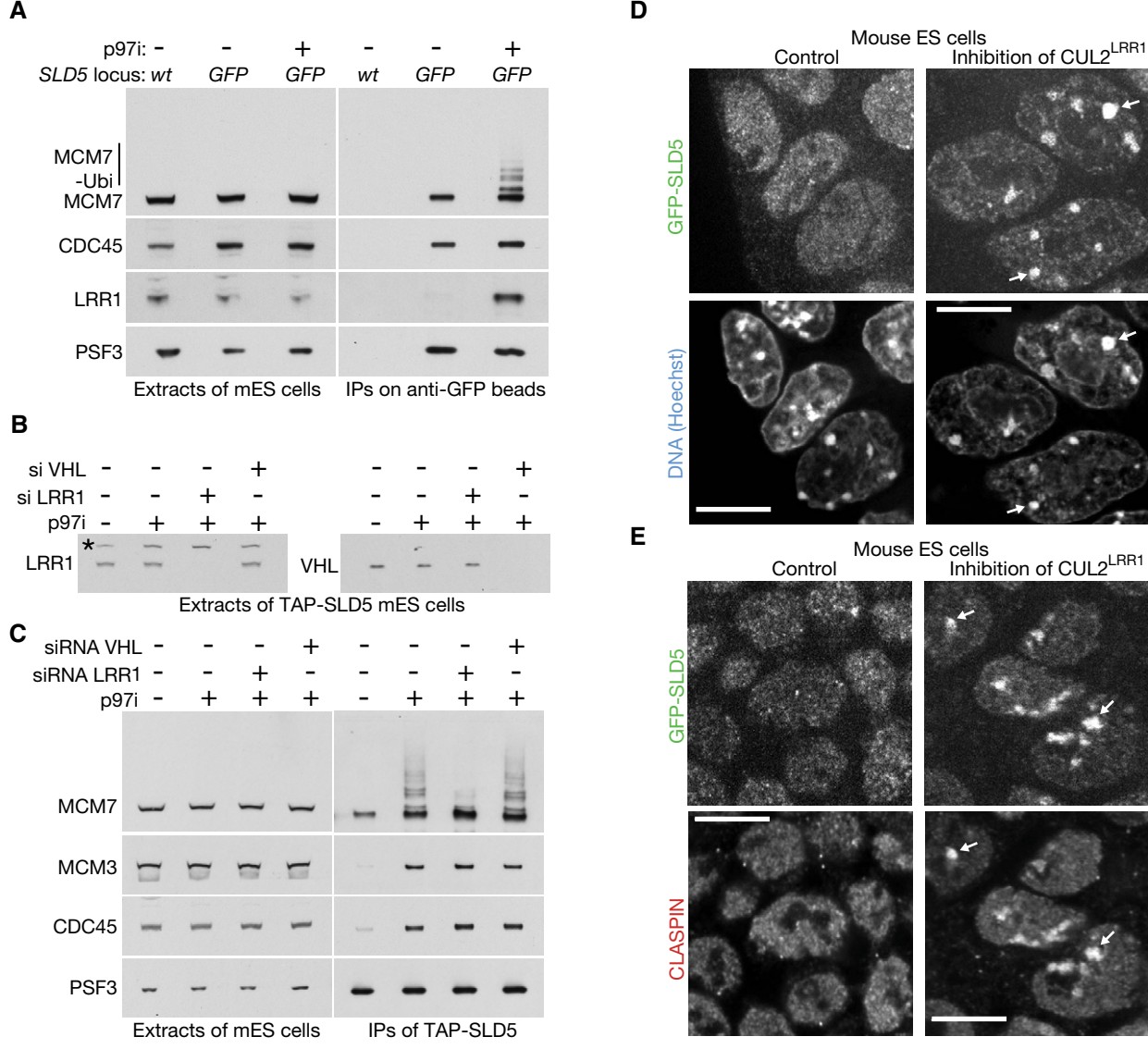

**Figure 3. CUL2^LRR1 is required for CMG ubiquitylation and replisome disassembly during S-phase in mouse ES cells.**

A    Cell extracts were generated after the indicated treatments and then incubated with anti-GFP beads. Factors co-purifying with GFP-SLD5 were monitored by immunoblotting.

B    TAP-SLD5 cells were subjected to the indicated treatments. The level of LRR1 and VHL was assessed by immunoblotting. The asterisk denotes a non-specific band.

C    For the experiment described in (B), TAP-SLD5 was isolated from cell extracts and the indicated factors monitored by immunoblotting.

D    The activity of CUL2^LRR1 was inhibited in GFP-SLD5 cells as described in Materials and Methods, by treatment with a combination of LRR1 siRNA and MLN4924 (the combined treatment was important to block CMG disassembly during DNA replication termination, and Fig EV2B shows combining LRR1 siRNA with MLN4924 produces a tighter block to CMG-MCM7 ubiquitylation than either individual treatment). Fixed cells were analysed as above. Arrows indicate the accumulation of GFP-SLD5 on constitutive heterochromatin (49% cells, *n* = 125).

E    Cells were treated as in (D) and the localisation of GFP-SLD5 was compared with CLASPIN by immunofluorescence. Arrows indicate the co-localisation of GFP-SLD5 and CLASPIN on constitutive heterochromatin (in 90% cells with GFP-SLD5 patches, *n* = 103).

Data information: Scale bars in all panels correspond to 10 μm.

After treatment for 3 h with p97 inhibitor, around half the cells contained brighter heterochromatin patches of GFP-SLD5 (Fig 2D). Moreover, such heterochromatin patches also contained other partners of CMG such as CLASPIN (Fig 2E(i)), CTF4 (Fig 2E(ii)) and POLE1 (Fig EV3C and D). Whereas mCherry-PCNA still associated transiently with heterochromatin patches after inhibition of p97, GFP-SLD5 persisted on chromatin after the disappearance of mCherry-PCNA (Fig EV3E). Given that CB-5083 caused CMG to accumulate with ubiquitylated MCM7 (Fig 2A), these findings indicated that p97 inhibition led to accumulation of ubiquitylated CMG on chromatin after DNA replication termination, reflecting an essential role for p97 in disassembly of the CMG helicase in mouse ES cells.

## CUL2^LRR1 promotes the ubiquitylation and chromatin extraction of the mammalian CMG replisome

Ubiquitylation of CMG-MCM7 was impaired when mouse ES cells were treated with the neddylation inhibitor MLN4924 (Soucy *et al*, 2009) in addition to p97 inhibitor (Fig EV2B). Since the principal role of neddylation is to activate the cullin family of ubiquitin ligases (Baek *et al*, 2020), this indicated that CMG-MCM7 ubiquitylation was dependent upon at least one member of this family. Subsequently, we found that the LRR1 substrate adaptor for Cullin 2 associated with the CMG helicase in cells treated with p97 inhibitor (Fig 3A), and depletion of LRR1 by siRNA impaired the ubiquitylation of CMG-MCM7 (Figs 3B and C, and EV2B and C; Appendix Fig S2A and B shows that two different siRNA to LRR1 produced a similar defect). This effect was specific to inactivation of CUL2^LRR1, since depletion of the alternative CUL2 adaptor VHL did not inhibit CMG-MCM7 ubiquitylation (Fig 3B and C).

Correspondingly, we found that inhibition of CUL2^LRR1 led to the accumulation of GFP-SLD5 and CLASPIN on constitutive heterochromatin (Fig 3D and E). Moreover, GFP-SLD5 persisted on heterochromatin patches after the disappearance of mCherry-PCNA (Fig 4A). Overall, therefore, these data indicated that CUL2^LRR1 is important for the ubiquitylation and chromatin extraction of the CMG replisome during DNA replication termination in mouse ES cells.

## TRAIP is required to extract the CMG helicase from chromatin during mitosis in mouse ES cells

When mouse ES cells lacking CUL2^LRR1 activity entered mitosis, the association of GFP-SLD5 with heterochromatin patches disappeared just before metaphase (Fig 4B and C). These findings indicated that mouse ES cells have a mitotic pathway for CMG helicase disassembly, which does not require CUL2^LRR1, analogous to the process of mitotic CMG disassembly that was originally observed in *C. elegans* early embryos and *X. laevis* egg extracts (Sonneville *et al*, 2017; Deng *et al*, 2019; Priego Moreno *et al*, 2019).

In a mass spectrometry analysis of the mammalian replisome to be described elsewhere (F.V., J.A. and K.L., unpublished data), we found that the TRAIP ubiquitin ligase associates with the CMG replisome in mouse ES cells (Fig 5A). Although TRAIP becomes essential for cell proliferation at an early stage of development in the mouse embryo (Park *et al*, 2007), we were able to introduce deletions into both copies of the TRAIP gene in mouse ES cells, which prevented

expression of functional TRAIP (Fig EV4A–E). TRAIP^−/− cells grew more slowly than control cells (Fig EV4F) and were sensitive to Mitomycin C that induces inter-strand DNA crosslinks (Fig EV4H).

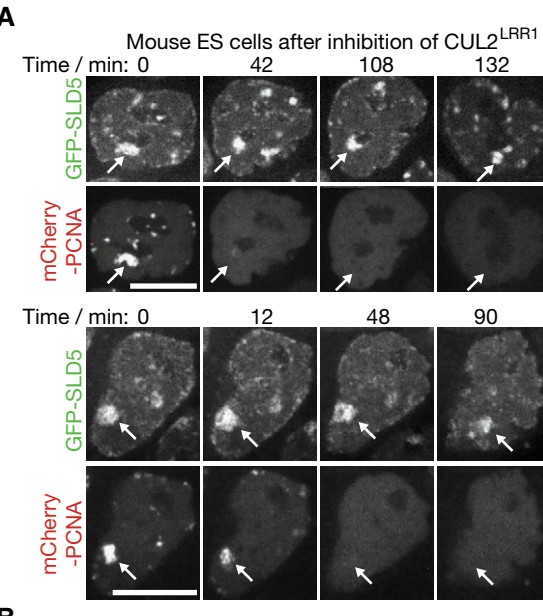

**A** Mouse ES cells after inhibition of CUL2^LRR1

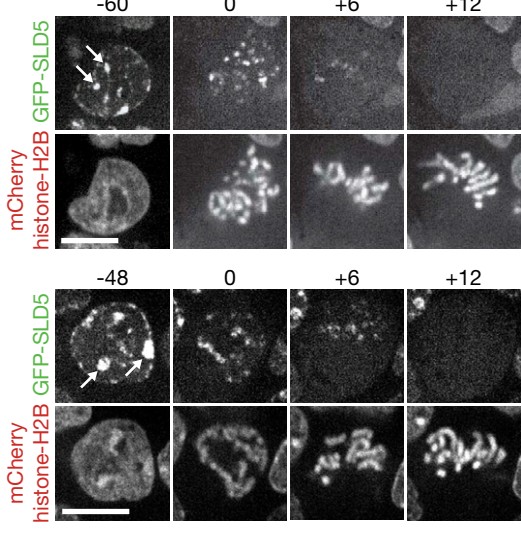

**B** Mouse ES cells after inhibition of CUL2^LRR1
Time / min (relative to Nuclear Envelope Breakdown)

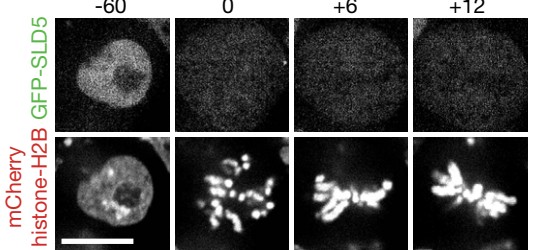

**C** Untreated mouse ES cells
Time / min (relative to Nuclear Envelope Breakdown)

**Figure 4. Upon inhibition of CUL2^LRR1, CMG persists on chromatin throughout interphase and is removed during early mitosis.**

A Time-lapse analysis of mouse ES cells expressing GFP-SLD5 and mCherry-PCNA, following inhibition of CUL2^LRR1. Arrows illustrate the persistence of GFP-SLD5 on heterochromatin after disappearance of mCherry-PCNA.

B Similar time-lapse analysis of mouse ES cells expressing GFP-SLD5 and mCherry-Histone H2B (from the *CAG* promoter at the *ROSA26* locus—see Materials and Methods), after inhibition of CUL2^LRR1. Arrows indicate heterochromatic patches of GFP-SLD5 that disappear during entry into mitosis (98% cells, *n* = 97).

C Equivalent data to (B) but for untreated cells (100% of untreated cells lacked GFP-SLD5 on mitotic chromatin, *n* = 149).

Data information: Scale bars in all panels correspond to 10 μm.

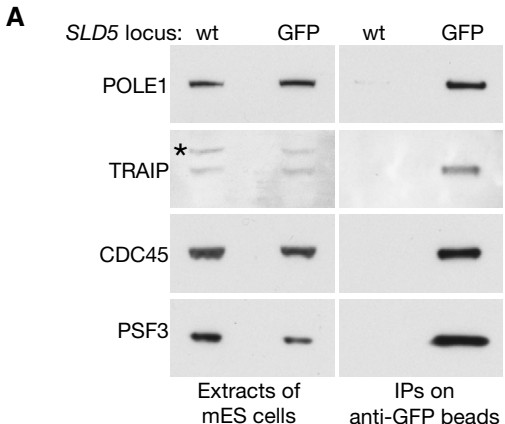

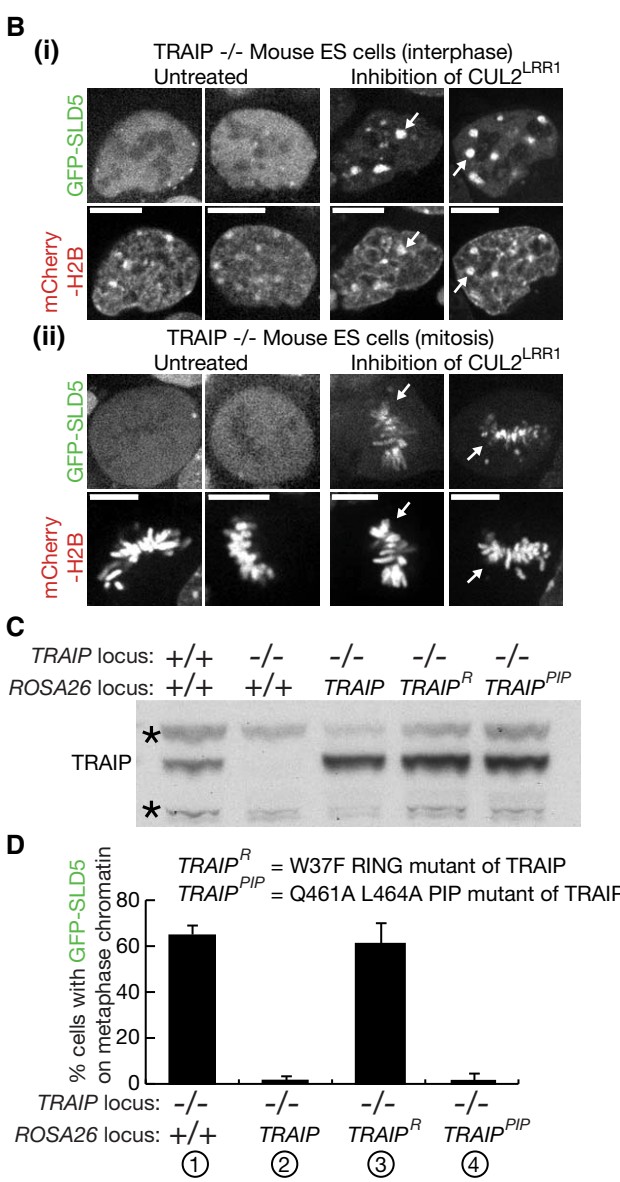

Figure 5. Mitotic CMG disassembly in mouse ES cells is dependent upon TRAIP E3 ligase activity but does not require association of TRAIP with PCNA.

A  TRAIP co-purifies with the CMG helicase from extracts of mouse ES cells. The asterisk indicates a non-specific band.
B  Mouse ES cells were processed as in Fig 3B. Arrows in (i) indicate accumulation of GFP-SLD5 on heterochromatin during interphase, whereas arrows in (ii) denote GFP-SLD5 on chromatin during mitotic metaphase. Scale bars correspond to 10 μm.
C  Immunoblot illustrating the stable expression from the CAG promoter of the indicated forms of TRAIP at the *ROSA26* locus in TRAIP$^{-/-}$ mouse ES cells. Asterisks indicate non-specific bands.
D  Quantification of the percentage of mitotic cells with GFP-SLD5 on mitotic chromosomes, after inhibition of CUL2$^{LRR1}$ as above, in cells with the indicated genotypes. Cells were fixed before analysis.

Data Information: In (D), data are presented as mean of three experiments ± SD.

These findings are consistent with the previously reported role of human TRAIP in DNA repair (Harley *et al*, 2016; Hoffmann *et al*, 2016; Soo Lee *et al*, 2016).

In untreated TRAIP$^{-/-}$ cells, GFP-SLD5 was not present on mitotic chromatin (Fig 5B (ii), left panels), reflecting the activity of CUL2$^{LRR1}$ during DNA replication termination. Upon inhibition of CUL2$^{LRR1}$ in TRAIP$^{-/-}$ cells, GFP-SLD5 accumulated on heterochromatin patches (Fig 5B (i), right panels), as described above for control cells (the lower proportion of TRAIP$^{-/-}$ cells with GFP-SLD5 patches, compared with wild-type cells, likely reflects the longer doubling time of TRAIP$^{-/-}$ cells, as shown in Fig EV4F). However, GFP-SLD5 remained on chromatin when TRAIP$^{-/-}$ cells lacking CUL2$^{LRR1}$ activity entered mitosis (Fig 5B (ii), right panels; Fig 5D sample 1). The defect in mitotic CMG disassembly was rescued by expression of wild-type TRAIP from the *ROSA26* locus, but not by expression of TRAIP with a mutation in the RING domain (Fig 5C and D) at a conserved site of interaction with E2 ubiquitin conjugating enzymes (Sarkar *et al*, 2019). Therefore, these data indicated that TRAIP ubiquitin ligase activity is required for the mitotic pathway of CMG disassembly in mouse ES cells. We also found that mutation of the PIP box at the C terminus of TRAIP did not impair mitotic CMG disassembly (Fig 5C and D). This indicated that TRAIP's role in the mitotic CMG disassembly pathway is independent of the PCNA interaction that was previously reported to recruit human TRAIP to sites of DNA damage (Hoffmann *et al*, 2016), and instead is likely to reflect the association of TRAIP with other components of the CMG replisome.

The conserved roles of CUL2$^{LRR1}$ and TRAIP in mouse ES cells, *C. elegans* and *X. laevis* indicate that metazoan species share two conserved pathways that lead to CMG helicase disassembly by p97 (Fig EV5). During S-phase, CUL2$^{LRR1}$ activates CMG helicase disassembly during DNA replication termination (Fig EV5A), analogous to the role of SCF$^{Dia2}$ in budding yeast. In contrast, TRAIP is required for a second pathway of CMG helicase disassembly during mitosis (Fig EV5B and C), which is likely to have evolved as the first step in the processing of sites of incomplete DNA replication (Bhowmick *et al*, 2016; Sonneville *et al*, 2019). A recent study of mitotic CMG disassembly in *Xenopus* egg extracts indicates that TRAIP-dependent CMG disassembly provides access to replication fork DNA for structure-specific nucleases that are activated during mitosis (Deng *et al*, 2019). Mitotic cleavage and repair of replication fork DNA allows chromosome segregation to proceed unhindered but is likely to come

at the price of small deletions in one of the two sister chromatids (Deng et al, 2019). This phenomenon likely explains the instability of "common fragile sites", which are a known feature of late replicating loci in mammalian cells (Glover et al, 2017), reflecting the difficulty of completing DNA replication before mitosis in cells with large genomes (Al Mamun et al, 2016; Moreno et al, 2016).

Much remains to be learnt regarding the action of the TRAIP ubiquitin ligase. TRAIP can associate with the mouse replisome during S-phase (Fig 5A), consistent with previous observations with extracts of *Xenopus laevis* eggs (Dewar et al, 2017; Deng et al, 2019; Priego Moreno et al, 2019). However, TRAIP only induces CMG disassembly upon entry into mitosis (this study; Deng et al, 2019; Priego Moreno et al, 2019) or when two forks converge at interstrand DNA crosslinks in Xenopus egg extracts (Wu et al, 2019). The molecular basis of this regulation remains to be determined.

Interestingly, it appears that neither CUL2$^{LRR1}$ nor TRAIP are found outside of metazoa, indicating that the regulation of CMG helicase disassembly has diversified to a surprising degree during the course of eukaryotic evolution. It will be important in future studies to explore how other eukaryotes regulate the final stages of chromosome duplication, in order to preserve genome integrity during the course of cell proliferation.

# Materials and Methods

Reagents and resources that were used in this study are listed in Appendix Table S1.

## Plasmid construction

PCRs were performed with the Phusion polymerase (M0530L, New England Biolabs), and DNA fragments were cloned via the Gibson Assembly Cloning Kit (E2611L, New England Biolabs) or with a combination of restriction enzymes and ligase. All new constructs were verified by Sanger sequencing. Site-specific mutagenesis was performed using the Phusion polymerase, according to the manufacturer's protocol.

## Maintenance of mouse ES cells

E14tg2a cells were cultured in a humidified atmosphere of 5% $CO_2$, 95% air at 37°C under feeder-free conditions with Leukemia Inhibitory Factor (LIF, DU1715, MRC PPU Reagents and Services) in serum-containing medium. Culturing dishes were precoated with 0.1% gelatin (G1890, Sigma) prior to seeding. Medium was based on Dulbecco's Modified Eagle Medium (DMEM, 11960044, Thermo Fisher), supplemented with 10% Foetal Bovine Serum (FBS, FCS-SA/500, LabTech), 5% KnockOut Serum Replacement (10828028, Thermo Fisher), 2 mM L-Glutamine (25030081, Thermo Fisher), 100 U/ml Penicillin-Streptomycin (15140122, Thermo Fisher), 1 mM Sodium Pyruvate (11360070, Thermo Fisher), a mixture of seven non-essential amino acids (11140050, Thermo Fisher; diluted to 1%), 0.05 mM β-mercaptoethanol (M6250, Sigma) and 0.1 μg/ml LIF. For passaging, cells were released from dishes using 0.05% Trypsin / EDTA (25300054, Thermo Fisher).

To determine doubling times, 2,000,000 cells (initial cell number N1) were grown for 48 h on a 10 cm plate, before recovery with 0.05% Trypsin / EDTA (25300054, Thermo Fisher). The cell suspension (final cell number N2) was mixed with an equal volume of Trypan blue stain (T10282, Invitrogen), before counting with a "Countess Automated Cell Counter" (C10227, Invitrogen, using Countess chamber slides, C10314, Invitrogen). The doubling time (G) was then calculated via the formula G = (48 × log(2)) / (log (N2) – log(N1)). The experiments were performed in triplicate and the mean values determined, together with the standard deviation (SD).

## Maintenance of human RPE1 cells

RPE1 cells (as confirmed by short tandem repeat profiling through ATCC) were cultured in a humidified atmosphere of 5% $CO_2$, 95% air at 37°C. The medium was based on Dulbecco's Modified Eagle Medium:Nutrion Mixture F-12 (21331020, Thermo Fisher), supplemented with 10% Foetal Bovine Serum (FBS, FCS-SA/500, LabTech), 2 mM L-Glutamine (25030081, Thermo Fisher) and 100 U/ml Penicillin-Streptomycin (15140122, Thermo Fisher). During passaging, cells were released from the dishes using 0.05% Trypsin / EDTA (25300054, Thermo Fisher).

## EdU incorporation to determine the percentage of S-phase cells

Cells were grown to 60–70% confluency in a 4-well chamber slide, before incubation with 10 μM 5-Ethynyl-2'-deoxyuridine (EdU) for 30 min. The cells were then fixed with 4% paraformaldehyde in phosphate-buffered saline (PBS) containing 1 mM $MgCl_2$ + 1 mM $CaCl_2$. The incorporated EdU was then labelled using the "Click-iT™ Plus Alexa Fluor™ 647 Picolyl Azide Toolkit" (C10643, Invitrogen), according to the manufacturer's instructions. Nuclear DNA was stained with 5 μg/ml Hoechst 33342 (H1399, Invitrogen) for at least 30 min, and then Hoechst and Alexa Fluor647 images were captured by spinning disk confocal microscopy (see below). The S-phase population was calculated by dividing the number of cells positive for EdU-Alexa Fluor647 by the total number of Hoechst-positive cells.

## Synchronisation of mouse ES cells and treatment with inhibitors of p97 or CUL2$^{LRR1}$

Cells were synchronised in early S-phase by treatment with 1.25 mM thymidine (T9250, Sigma) for 18 h and then released for at least 7 h in fresh medium containing 5 μM STLC (164739, Sigma), in order to arrest cells in mitosis. Subsequently, cells were washed three times with PBS and released into fresh medium. The quality of cell-cycle synchronisation was monitored by flow cytometry analysis.

To inhibit the p97 ATPase, cells were treated with 5 μM CB-5083 (S8101, Selleckchem) for 3 h before harvesting. To inhibit the activity of cullin ligases, 5 μM MLN4924 (A-1139, Activebiochem) was added to the culture medium for 5 h before harvesting (except for the experiment in Fig 5D, in which MLN4924 was added for 12 h before cells were fixed).

## CRISPR-Cas9 genome editing

To modify a specific site in the mouse genome, a pair of guide RNAs (gRNAs) were designed (https://wge.stemcell.sanger.ac.uk//find_c

risprs), in order to direct cleavage of both parental DNA strands by a "humanised" version of the *S. pyogenes* Cas9 "nickase" (Cas9-D10A). Each gRNA was based on 20 nt of homology to a target sequence in the mouse genome, which was located immediately upstream of a 3-nt "Protospacer Adjacent Motif" or PAM sequence of the form "NGG" (examples are shown in Figs 1A and EV4D). In each case, two annealed oligonucleotides containing the 20 nt homology region (see Appendix Table S1) were cloned as previously described (Pyzocha *et al*, 2014) into the vectors pX335, pKN7 or pKN101 (see Appendix Table S1), after cleavage with the type II S restriction enzyme BbsI.

To create small deletions at a particular locus, mouse ES cells were transfected as described below, with two plasmids expressing the chosen pair of gRNAs together with Cas9-D10A and the puromycin resistance gene (Fig EV4A). To introduce enhanced GFP (eGFP) or the TAP tag into the SLD5 locus (Fig 1D and E), or to introduce plasmids expressing TRAIP or mCherry-HistoneH2B into the *ROSA26* locus (Figs 4B and C, 5B–D, and EV4G), a further plasmid containing donor DNA was also included in the transfection. The donor DNA included around 0.5–1 kb of flanking homology to either side of the target locus (Appendix Table S1). Care was taken to ensure that neither of the two gRNAs would be able to anneal with the target locus after successful genome editing (e.g. see Fig 1B). Note that knockins with Cas9 nickase were performed with either one gRNA (integration of TRAIP into *ROSA26* locus) or two gRNAs (all other experiments in this study).

### Random integration of a plasmid expressing mCherry-PCNA

mCherry-PCNA was expressed from the CMV promoter, via random integration of a linearised plasmid (pcDNA3.1-mCherry-PCNA, see Appendix Table S1) that also contained a kanamycin / G418 resistance gene. Transfected cells were selected for 9 days with medium containing 300 µg/ml G418 (G418S, Formedium), before single-cell sorting via flow cytometry. Stable clones were then monitored by immunoblotting and spinning disk confocal microscopy.

### Transfection of mouse ES cells and selection of clones

A stock solution of 1 mg/ml linear Polyethylenimine (PEI, 24765-2, Polysciences, Inc) was prepared in a buffer containing 25 mM HEPES, 140 mM NaCl, 1.5 mM $Na_2HPO_4$ adjusted to pH 7.0 with NaOH. The solution was sterilised by passing through a 0.2 µm filter and stored at − 20°C. Subsequently, $1 \times 10^6$ cells were aliquoted and centrifuged at 350 g for 5 min, before resuspension in 100 µl DMEM (without supplements) and mixed by pipetting. Next, 1 µg of each plasmid DNA (two plasmids expressing Cas9 and gRNA(s), plus donor vector as appropriate) was added and the cells were mixed by gentle pipetting, before addition of 15 µl PEI. After further gentle mixing, the cells were incubated at room temperature for 30 min, before transfer to a single well of a 6-well plate that had been precoated with 0.1% gelatin (G1890, Sigma) and contained 2 ml of complete DMEM medium (supplements as above). Cells were incubated for 24 h after transfection, before two 24-h rounds of selection with fresh medium containing 2 µg/ml Puromycin (A1113802, Thermo Fisher). The surviving cells were then released from the well with 0.05% Trypsin / EDTA (25300054, Thermo Fisher). Flow cytometry was then used to sort single cells into

200 µl supplemented DMEM medium in individual wells of a 96-well plate, which had been precoated with 0.1% gelatin (G1890, Sigma). Cells colonies were observed ∼ 14 days later and any viable clones were expanded and then monitored as appropriate by immunoblotting, PCR and DNA sequencing of the target locus.

### RNA interference

Lipofectamine RNAiMAX (13778, Thermo Fisher) was used to introduce siRNA into mouse ES cells, according to the manufacturer's protocol. The LRR1 and VHL siRNAs and control siRNA were transfected at a final concentration of 25 nM. After 24 h, the transfection was repeated and incubation continued for a further 24 h.

Initially, four individual siRNAs for LRR1 were tested (LQ-057816-01, Horizon Discovery), in order to identify the most effective (J-057816-10, Horizon Discovery). The latter was then used for all experiments in this manuscript, except for the experiment in Appendix Fig S2 that also employed a second LRR1 siRNA (J-057816-09, Horizon Discovery). Sequence details for both LRR1 siRNA are provided in Appendix Table S1. In the case of VHL, a "Smart Pool" of four different siRNAs was employed (L-040755-00, Horizon Discovery).

### PCR and sequence analysis for genotyping of mouse ES cells

Cells from a single well of a 6-well plate were resuspended in 50 µl of $dH_2O$, then 50 µl of 100 mM NaOH was added and the sample was incubated at 95°C for 15 min. Subsequently, 11 µl of 1 M Tris–HCl (pH 7.0) was added with mixing. Finally, 1 µl was used as the genomic DNA template for genotyping purposes, in 20 µl PCRs, using the oligos indicated in Fig EV4B and listed in Appendix Table S1. PCR products from TRAIPΔ clones were cloned and sequenced, confirming the presence of small deletions in the region targeted by Cas9-D10A. Examples are shown in Fig EV4D.

### Extracts of mouse ES cells and immunoprecipitation of protein complexes

For all such experiments except for Fig 3A, three to five 15 cm petri dishes were precoated with 0.1% gelatin (G1890, Sigma) and seeded with $7.5 \times 10^6$ mouse ES cells per plate, which were then grown for 48 h at 37°C. Cells were then washed with PBS and released from the dishes by incubation for 10 min with PBS containing 1 mM EGTA and 1 mM EDTA, before harvesting by centrifugation. Typically, 0.5–0.8 g of cell pellet was obtained for each sample. The remaining steps were performed at 4°C and are based on our previously described methods for isolating protein complexes from yeast cells (Gambus *et al*, 2006; Maric *et al*, 2014). Cells were resuspended with one volume of lysis buffer (100 mM HEPES-KOH pH 7.9, 100 mM potassium acetate, 10 mM magnesium acetate, 2 mM EDTA, 10% glycerol, 0.1% Triton X-100), supplemented with 2 mM sodium fluoride, 2 mM sodium β-glycerophosphate pentahydrate, 1 mM dithiothreitol (DTT) and 1% Protease Inhibitor Cocktail (P8215, Sigma-Aldrich). Deubiquitylase enzymes were inhibited by the addition of 5 µM Propargyl-Ubiquitin (DU49003, MRC PPU Reagents and Services; Ekkebus *et al*, 2013), and chromosomal DNA was digested for 30 min at 4°C with 1,750 U of Pierce Universal Nuclease (88702, Thermo Fisher). The extracts were centrifuged at 20,000 g for 30 min at 4°C and a 50 µl aliquot of the supernatant

was added to 100 µl of 1.5 × Laemmli buffer and stored at −80°C for immunoblotting.

The remainder of each extract was then incubated for 120 min with 40 µl of GFP-Trap Agarose beads (gta-100, Chromotek) that had been pre-washed in PBS supplemented with 5 mg/ml BSA, or with $1.7 × 10^9$ Dynabeads M-270 Epoxy (14302D, Thermo Fisher) that had been coupled to rabbit immunoglobulin G (S1265, Sigma-Aldrich). The beads were washed four times with 1 ml of wash buffer (100 mM HEPES-KOH pH 7.9, 100 mM potassium acetate, 10 mM magnesium acetate, 2 mM EDTA, 0.1% IGEPAL CA-630, 2 mM sodium fluoride, 2 mM sodium β-glycerophosphate pentahydrate, plus protease inhibitors as above) and bound proteins were eluted at 95°C for 5 min in 100 µl of 1 × Laemmli buffer and stored at −80°C for subsequent analysis.

The samples in Fig 3A were processed as above, except that ten 15 cm petri dishes were used per sample, and GFP-SLD5 was isolated with 50 µl of GFP-Trap Agarose beads. The bound proteins were then eluted with 30 µl Laemmli buffer.

### Antibodies and immunoblotting

New antibodies produced in this study are listed in Appendix Table S1 and validated in Figs 3B, 5C, EV2C, and EV4E and Appendix Fig S3. Protein samples were resolved on NuPAGE Novex 4–12% Midi Bis-Tris gels (NP0301, Thermo Fisher) with NuPAGE MOPS SDS buffer (NP000102, Thermo Fisher). The resolved proteins were transferred onto a nitrocellulose membrane (IB301031, Thermo Fisher) using the iBlot Dry Transfer System (Thermo Fisher).

### Immunofluorescence and spinning disk confocal microscopy

For immunofluorescence analysis, cells were grown on chambered 4-well slides (µ-slide 4 well, 80426, Ibidi) and then fixed in 4% (w/v) paraformaldehyde in PBS for 10 min at room temperature. Cell membranes were permeabilised with PBS supplemented with 3% BSA, 1 mM MgCl₂, 1 mM CaCl₂ and 0.1% Triton X-100 for 10 min. After blocking with PBS supplemented with 3% BSA, 1 mM MgCl₂, 1 mM CaCl₂ for 30 min, primary antibodies were diluted 1:100 in PBS supplemented with 3% BSA, 1 mM MgCl₂ and 1 mM CaCl₂, and incubations were performed at 4°C overnight. The chamber slides were then washed three times with 3% BSA, 1 mM MgCl₂ and 1 mM CaCl₂.

Secondary antibodies were diluted 1:1,000 in PBS supplemented with 3% BSA, 1 mM MgCl₂, 1 mM CaCl₂ and 1 µg/ml Hoechst 33342 (to stain DNA), and incubations were performed at room temperature in the dark for 60 min. After washing three times, the chamber slides were mounted with 700 µl of PBS containing 1 mM MgCl₂ and 1 mM CaCl₂.

Confocal images of fixed or live cells were acquired with a Zeiss Cell Observer SD microscope with a Yokogawa CSU-X1 spinning disk, using a HAMAMATSU C13440 camera with a PECON incubator, a 60 × or 100 × 1.4-NA Plan-Apochromat oil-immersion objective, and appropriate excitation and emission filter sets for up to three different wavelengths. Images were acquired using the "ZEN blue" software (Zeiss) and processed with ImageJ software (National Institutes of Health) as previously described (Sonneville *et al*, 2017).

For all samples in a particular experiment, the conditions for image capture were identical and the data were processed in the same way.

### Monitoring sensitivity of mouse ES cells to Mitomycin C

For each sample, $4$–$8 × 10^5$ cells were plated into a single well of a 24-well plate. After 6 h, the medium was changed for fresh medium containing 0–100 ng/ml Mitomycin C and incubation was continued for 24 h. The drug-containing medium was then replaced with fresh medium lacking Mitomycin C, and incubation proceeded for a further 4 days. The cells were then washed with PBS and fixed with methanol. The fixed cells were stained with 0.5% crystal violet solution (HT90132, Sigma-Aldrich), and images of the plates were then captured with a scanner.

### Flow cytometry

Cells were released from petri dishes as above and then fixed with 1 ml 70% ethanol, added dropwise whilst vortexing the cells. For each sample, 100 µl of fixed cells was washed in PBS and then resuspended in PBS containing 0.05 mg/ml propidium iodide (P4170 Sigma-Aldrich) and 0.05 mg/ml RNase A (Sigma-Aldrich). Cells were incubated at 25°C for 1 h, and the samples were then processed using a FACSCanto II flow cytometer (Becton Dickinson) and the data were analysed using FlowJo software (TreeStar).

### USP2 deubiquitylation assay

For the experiment in Fig EV2A, ubiquitylated CMG was isolated as described above by immunoprecipitation of TAP-SLD5 from mouse ES cells treated with 5 µM CB-5083. The purified complexes were then treated for 1 h at 24°C with 1 µM Hs USP2 (DU13025, MRC PPU Reagents and Services) in 100 mM HEPES-KOH pH 7.9, 100 mM potassium acetate, 10 mM magnesium acetate, 0.02% (v/v) IGEPAL CA-630, 1 mM DTT, with agitation at 1,400 rpm on an Eppendorf ThermoMixer F1.5. The reactions were stopped by addition of 20 µl of 3 × Laemmli buffer and were then boiled for 5 min at 95°C, before analysis by immunoblotting.

### Statistics and reproducibility

The experiments in Figs 2A and C–E, 3A and D–E, 4A–C, 5A and D, EV2B and C, and EV3A,D and E were performed three or more times. The experiments in Figs 1G–J, 3B and C, 5B, EV3B and C, and EV4H were carried out twice. The experiments in Figs 1F, 2B, 5C, EV1, EV2A, and EV4C, E and F, and Appendix Figs S1–S3 were performed once.

# Data availability

No primary data sets have been generated and deposited.

Expanded View for this article is available online.

### Acknowledgements

We thank Remi Sonneville, Iain Porter and Paul Appleton for invaluable assistance with spinning disk confocal microscopy, Masato Kanemaki for a plasmid expressing mCherry-PCNA and MRC PPU Reagents and Services (https://mrc ppureagents.dundee.ac.uk) for antibody production. We are very grateful for financial support from the Medical Research Council (core grant

MC_UU_12016/13 to KL), the Wellcome Trust (reference 102943/Z/13/Z for an Investigator award to KL), Cancer Research UK (Programme Grant C578/A24558 and PhD studentship C578/A25669 to KL; Core Funding C5759/A20971 to GL), Biotechnology and Biological Sciences Research Council (reference BB/I001794/1/BB for award to GL), the Japan Society for the Promotion of Science (postdoctoral fellowship and "Kakenhi" grants JP19K06611 and JP20K21423 to KN) and the Naito Foundation (postdoctoral fellowship to KN).

## Author contributions

Experiments were performed by FV (Figs 1F–J, 2A, 3B and C, 5A, EV1, and EV2A–C, Appendix Figs S1A, S2 and S3), RF (Figs 2B–E, 3D–E, 4A–C, 5B–D, EV3A–E, and EV4A–H and Appendix Fig S1B), JA (Fig 3A) and KN (initial establishment of CRISPR-Cas9 in mouse ES cells and Fig 1A–E). MLAL and GL taught KN to work with mouse ES cells. The project was designed by FV, RF, KN and KL. KL wrote the manuscript with critical input from the other authors.

## Conflict of interest

The authors declare that they have no conflict of interest.

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
