## [Review Process File · EMBO Reports]

CUL2^{LRR1}, TRAIIP and p97 control CMG helicase disassembly in the mammalian cell cycle

Fabrizio Villa, Ryo Fujisawa, Johanna Ainsworth, Kohei Nishimura, Michael Lie-A-Ling, Georges Lacaud, and Karim Labib

DOI: [10.15252/embr.202052164](https://doi.org/10.15252/embr.202052164)

Corresponding author(s): Karim Labib (kpmlabib@dundee.ac.uk)

Review Timeline:	Transfer from Review Commons:	24th Nov 20
	Editorial Decision:	7th Dec 20
	Revision Received:	16th Dec 20
	Accepted:	7th Jan 21

Editor: Esther Schnapp

The logo for Review Commons, with the word "Review" in a large, blue, italicized serif font and the word "COMMONS" in a smaller, blue, uppercase sans-serif font below it.

Transaction Report: This manuscript was transferred to EMBO Reports following peer review at Review Commons

**RESPONSE TO REVIEWERS:**

We are grateful to the three reviewers for their comments and helpful
suggestions. All three reviewers were united in thinking that our experiments were
**“of impeccable technical quality”** and felt **“that the manuscript is publishable**
**without or with only minor revisions”**, since the **“main conclusions are**
**sufficiently supported by experimental evidence”**.

In order to address all of the points that were raised by the reviewers, we
have now submitted a revised manuscript that includes:

- - a considerable amount of new data (Figure 2C, Figure 4C, Figure EV1, Figure
 - EV4F, Appendix Figure 1, Appendix Figure 2, Appendix Figure 3)
 - - new Figures (Figure EV5)
 - - a substantial revision of the text to address all of the points that were raised.
 - - rearrangement of previous figures (largely in response to the single major point that
 - was raised by Reviewer 1). Therefore, the revised manuscript now has 5 main
 - figures, 5 ‘Extended View’ figures, 3 ‘Appendix Supplementary Figures’ and 1
 - Appendix Supplementary Table.

For all these reasons, we hope that you will agree that our revised manuscript
is now ready for publication.

**1. SUMMARY OF REVIEWERS’ COMMENTS:**

**Reviewer 1:**

**The reviewer noted “the authors address a timely and very focused**
**research question in connection with the disassembly of the replisome.” “The**
**authors of the present study elegantly and convincingly transfer the insight**
**gained in other model systems to mammals and, thus, show that the**
**Cul2_LRR1 and TRAIIP pathways are required for CMG disassembly in**
**mammalian cells.”**

**“The established method for CMG isolation could indeed be of high**
**value to characterise the mammalian replisome and its fate in diverse**
**conditions that involve, for example, replisome stalling, similarly to an**
**equivalent method the lab developed using budding yeast, and has used for**
**years to gain insight into the structure, dynamics and working principles of the**
**eukaryotic replisome”**

**Reviewer 2**

**“the introduction of mouse ES cells as a tractable system accompanied by**
**gene editing method in order to study mammalian replisomes in spontaneous**
**and drug-treated conditions will be the major contribution since it can be a**
**valuable tool kit to the future research in studying replisomes in mammalian**
**cells. I believe this manuscript be publishable in your journal after some minor**
**revisions.”**

**Reviewer 3:**

**“The methods developed here with mESCs also open the road for future**
**analyses of the different proteins involved in this process.”**

**2. DETAILED REPLY TO REVIEWERS’ COMMENTS**

In the following discussion, please note that references to page and line
numbers correspond to the revised manuscript that we have submitted with tracked
changes. We have also submitted a further 'Related Manuscript File' containing the
revised version of all the figures and supplementary information.

REVIEWER 1

Major comments:

The reviewer summarised her / his view as follows: ***"The experiments
presented are of impeccable technical quality. The main conclusions are
sufficiently supported by experimental evidence. The research is presented in
an appropriate and balanced way."***

and then raised one main point:

***"The use of mouse ES cells for the presented study needs further discussion
and, perhaps, some development. The authors state that ES cells are perfectly
suitable for isolating CMG, because 60 % of each population are in S phase. I
agree that this might help, but more conventional lines have not much fewer
cells in S phase. Mouse ES cells are difficult to cultivate and manipulate in
high quantities on feeder cells or on gelatine with LIF in a way that they stably
retain pluripotency. However, the particular advantages of ES cells are not
discussed in the manuscript. Do the authors want to explore any of these
advantages? For example, do they want to specifically investigate how
replication forks in cells with embryonic cell cycles behave? Or do they want
to investigate how replisomes change during differentiation? Because the
authors make a point that the development of the experimental system of CMG
isolation is part of the scientific progress presented in the manuscript, a better
discussion of the matter is required. If specific attributes of ES cells are indeed
important, like pluripotency, differentiation capacity, stem cell-ness or
embryonic cell cycle and replication profiles the authors should characterise
whether the developed ES cell line serves the purpose."***

E14TG2a mouse ES cells have a stable diploid karyotype, in common with
stable diploid human cell lines such as RPE1. In practice, however, it is much
easier to isolate the CMG helicase from mouse ES cells, for multiple reasons:

- previous studies indicated that around 70% of asynchronously growing mouse ES
cells are in S-phase (Ter Huurne et al, now cited on lines 200-201), compared to
about 25-35% of human RPE1 cells (Matson et al, 2017, cited on lines 202-203).
Since this is an important point, we performed flow cytometry and EdU labelling of
E14TG2a mouse ES cells and human RPE1 cells grown in parallel with each other,
and found that the mouse ES cells had 72% S-phase cells, compared to 37% in
human RPE1 cells (the new data are now shown in Appendix Figure S1 and are
discussed between lines 200-202).

- mouse ES cells grow very rapidly, with a doubling time of 12.5 hour (Figure EV4F),
compared to a doubling time of over 20 hours for human RPE1 cells (Matson et al,
2017, cited on lines 208-209).

- mouse ES cells are smaller than somatic cells such as RPE1 cells and lack contact
inhibition (discussed in Burdon et al, 2002, which we now cite on line 206). For

these reasons, mouse ES cells can be grown at much higher densities per plate than
human RPE1 cells.

The reviewer was also concerned that mouse ES cells are difficult to cultivate,
but in reality the E14TG2a mouse ES cells used in our study are extremely easy to
grow and maintain in the absence of feeder cells, according to the protocols
described in Materials and Methods (we now note this point at the start of Results on
line 191-192).

Furthermore, mouse ES cells have large sub-nuclear bodies of constitutive
heterochromatin (Saksouk et al, 2015), which enabled us to develop a new in situ
method for monitoring microscopically the presence of the CMG-replisome on
replicating chromatin in live cells. We now note this point in Results on lines 210-
213.

In addition, genome editing in mouse ES cells is even easier than in human
diploid cell lines, allowing us to generate knockin lines without needing to insert a
marker gene into the modified locus (e.g. data now in Figure 1, with the details
described in Materials and Methods).

For all these reasons, E14TG2a mouse ES cells provide an ideal model
system with which to isolate the mammalian CMG helicase and study its regulation.

**Minor points to address:**

1. ***“Establishment of CMG isolation:***

***S phase specific, DNA-dependent Cdc45 and Mcm2-7 co-purification with Sld5***
***is very strong evidence that CMGs from replisomes are isolated.***

***However, the proteins involved might form diverse protein complexes, which***
***makes the simple statement that CMG is isolated difficult. Showing***

***dependency on replication initiation, for example by RNAi against origin***

***licensing or firing factors (Cdc6, Cdt1, Treslin/TICRR, MTBP) would***

***complement the experiments shown. Alternatively, co-purification of other***

***replisome components could be tested. Pol epsilon is shown, but, as an origin***
***firing factor, may not only interact with mature CMGs.***

***IP showing other replisome proteins”***

We agree with the reviewer that the S-phase specific and DNA-dependent co-

purification of SLD5 with all of the other 10 subunits of the CMG helicase (e.g.

Figures 1-2) provides very strong evidence that we are indeed isolating the CMG

helicase in our experiments.

We now present additional new data in Figure EV1, to confirm that the purified

material also contains the CMG partners TIMELESS-TIPIN, CTF4 and CLASPIN,

together with Pol alpha (POLA1), complementing data in Figure 5A that show co-

purification with Pol epsilon (POLE1).

These data strongly indicate that E14TG2a mouse ES cells provide a

powerful model system with which to isolate and characterise the mammalian CMG-

replisome.

2: ***"GFP-Sld5 microscopy:***
***The authors take co-localisation of Sld5 with PCNA in heterochromatic regions***
***as evidence that the Sld5 signal represents CMG in replisomes. I agree that***
***this is highly suggestive. Additional evidence that other replisome***
***components are also present erases almost all doubts. Because presented are***
***mere correlations, active manipulation of replisomes to preserve them on***
***heterochromatin that should therefore prevent termination and Sld5 unloading***
***could complement these correlations. Such a treatment could be replisome***
***stalling by high concentrations of aphidicolin or HU (perhaps in combination***
***with DDK inhibition to prevent dormant origin firing)."***

The reviewer notes that the colocalization of SLD5 with PCNA in
heterochromatic regions ***"is highly suggestive"*** of the presence on chromatin of the
CMG helicase at such sites. Moreover, ***"additional evidence that other replisome***
***components are also present erases almost all doubts"***.

Nevertheless, the reviewer suggested that the correlative data could be
complemented by ***"active manipulation of replisomes to preserve them on***
***heterochromatin"***.

In fact, our manuscript already contained such data, in the experiments where
we blocked replisome disassembly during S-phase by inhibition of p97 (data now in
Figure 2D-E) or by inhibition of CUL2-LRR1 (data now in Figure 3D-E and Figure
4A). In contrast, treating cells with aphidicolin or HU would have blocked late origin
firing via the S-phase checkpoint and so would not have been suitable for examining
the replisome on late-replicating heterochromatin.

Most importantly, the data now in Figure EV3E show that GFP-SLD5 and
mCherry-PCNA arrive at heterochromatin patches with the same kinetics in cells
treated with p97 inhibitor (Figure EV3E, top cell, compare t-18 and t0), but mCherry-
PCNA then disappears quickly as in untreated cells, whereas GFP-SLD5 remains on
chromatin. These data reflect the essential role of p97 in CMG helicase disassembly
during DNA replication termination and provide strong evidence to confirm that the
GFP-SLD5 signal on chromatin represents CMG in replisomes.

3. ***"Fig 1:***

***It is believably shown that p97i results in ubiquitylation of Mcm7 in CMG***
***isolations and suppression of CMG extraction in heterochromatic regions.***

***- The fact that Mcm7 present in CMGs is ubiquitylated in response to p97i, and***
***in light of what we know about disassembly of CMG in Xenopus, suggest that***
***ubiquitylation occurs specifically on CMGs. However, p97i could theoretically***
***also lead to ubiquitylation of pre-RCs or soluble Mcm7. Testing bulk chromatin***
***from G1 and S phase cells and soluble Mcm7 could complement the***
***experiments shown."***

The data in our manuscript show that 'soluble MCM7' is not detectably
ubiquitylated upon treatment of cells with p97 inhibitor, in contrast to the small
fraction of MCM7 in the CMG helicase (e.g. data now in Figure 2A, top panels:
compare free MCM7 in lane 3 with CMG-MCM7 in lane 6).

Although we agree with the reviewer that it would be interesting to explore
whether pre-RCs are ubiquitylated upon inhibition of p97, our manuscript is focussed
on the regulation of the CMG helicase, so that point remains beyond the remit of our
study.

- ***“Fig 1D: The image shown for mES cells (control) looks like it has a different***
***signal-to-noise ratio than the other images of this panel. The same is true for***
***at least one more figure, 2D. Can the authors comment whether all images***
***were captured and processed equally or, if not, give the details and explain?”***

We have now reprocessed the relevant data previously in Figure 1D and
Figure 2D (now 2D and 3D), and have adjusted the text in Materials and Methods
(lines 709-710) to confirm that ***“For all samples in a particular experiment, the***
***conditions for image capture were identical and the data were processed in the***
***same way.”***

- ***“Fig 1D: A more complete way of data quantification should be used to go***
***with this figure. In the text the authors write from line 200: “After treatment for***
***3 hours, around half the cells contained heterochromatin patches with GFP-***
***Sld5”...***

***What is the percentage without p97i or at 0 h?”***

As discussed on line 273, we have now quantified the percentage of
untreated cells that have PCNA / GFP-SLD5 on heterochromatic patches (14 %).

As illustrated in Figure 2C-D (previously Figure 1C-D), and discussed on lines
279-281, the heterochromatin patches of SLD5 are brighter after inhibition of p97.
This is likely due to the accumulation of ubiquitylated CMG on chromatin after DNA
replication termination, in the many replicons within each heterochromatin patch.

***“Are half the cells in late S at this point in time or do Sld5-positive***
***heterochromatic regions accumulate over time because CMGs are not***
***unloaded (which could be expected)?”***

The data in Figure EV3E (previously Figure S4E) show that mCherry-PCNA
associates transiently with heterochromatic patches in cells treated with p97i, just
like in control cells. GFP-SLD5 arrives on heterochromatin patches with the same
kinetics as mCherry-PCNA in cells treated with p97 inhibitor (Figure S3E, top cell,
compare t-18 and t0), but GFP-SLD5 remains on chromatin after the disappearance
of mCherry-PCNA. Therefore, these data indicate that the accumulation of cells with
GFP-SLD5 on heterochromatin reflects the fact that CMG is not unloaded during
DNA replication termination, in cells that lack p97 activity.

4. ***“Fig 2:***

***The conclusion that Cul2_LRR1 is required for Mcm7 ubiquitylation and to***
***remove CMG from heterochromatin is largely convincing.***

- ***Fig2B/C: A second siRNA against LRR (or a rescue with siRNA-resistant***
***LRR1) should be shown to exclude off-target effects.”***

New data in Appendix Figure S2 show that two different siRNA to LRR1
produce a comparable defect in CMG-MCM7 ubiquitylation. These data are now
discussed on lines 321-322.

***“Fig 2D/E: It seems that the authors use LRR1 siRNA + MLN4924 to inactivate***
***Cul2-LRR1. Please comment on whether individual treatments were not***
***effective enough or whether there is another reason.”***

The data now in Figure EV2B (previously Figure S3B) show that the
combination of MLN4924 and LRR1 siRNA produces a tighter block to CMG-MCM7
ubiquitylation than either individual treatment. MLN4924 inhibits the E1 enzyme for
neddylation and should in theory inhibit all cullin ligases. Since the phenotype of

MLN4924 is made tighter by LRR1 siRNA, this indicates that both of the individual
treatments are a bit leaky.

Correspondingly, the accumulation of replisome proteins on heterochromatin
patches in Figure 3D-E (previously Figure 2D-E) was dependent upon the combined
treatment of cells with MLN4924 and LRR1 siRNA. This is now explained in the
legend to Figure 3 (lines 2750-2755).

5. **"Fig 3:**

- **3A: This experiment seems to be missing a comparison with PCNA and Sld5**
**dynamics in untreated cells. A time lapse experiment with untreated cells is**
**shown in 1C. It seems to indicate that PCNA and Sld5 leave heterochromatin**
**without a delay. However, without a common reference time point (common 0**
**min time point) and without similar time points shown comparison is difficult.**
**For example, the authors could make a statement about whether there is a**
**delay between PCNA and Sld5 and how big it is."**

Careful analysis of time-lapse data (an example is provided in Figure 2C,
previously Figure 1C) shows that there is no delay between the arrival on
heterochromatin patches of mCherry-PCNA and GFP-SLD5. Moreover, the two
proteins disappear from heterochromatin patches with identical kinetics.

The revised text now makes this clear (lines 272-275), by saying that **"GFP-**
**SLD5 was readily detected on heterochromatic patches during late S-phase**
**(14% asynchronous cells, n = 238), appearing and disappearing with similar**
**kinetics to mCherry-PCNA (100% cells, n = 101; an example is shown in Figure**
**2C)."**

- **"3B: The authors should show that there is no Sld5 on mitotic chromosomes**
**in cells without Cul2 inhibition. This is required to unequivocally show that the**
**Sld5 signals seen on mitotic chromosomes are from CMGs not unloaded in the**
**previous S phase. In Fig 4 the authors show this for TRAI^{-/-} cells. A reference**
**to this fact may suffice."**

The revised text now makes clear on lines 347-349 that GFP-SLD5 is never
observed on mitotic chromatin in untreated cells. Moreover, examples of untreated
cells entering mitosis are shown in Figure 4C.

6. **"Fig 4: - 4C/D:**

**I do not think the authors state in the main text, legend or methods how they**
**complemented the cells. By transient transfections, random integration, using**
**plasmids or BACs?"**

Sorry about that – this was actually explained in Figure S5F (now Figure
EV4G), but for greater clarity, the relevant details have now been added to the
legend to Figure 5C (lines 2787-2788) and also to the appropriate section of
Materials and Methods (lines 541-550). Plasmids expressing TRAI^{-/-} from the CAG
promoter were integrated at the ROSA26 locus via CRISPR-Cas9 genome editing.

7. **"line 266:**

**"38 % cells had GFP-Sld5..." on heterochromatin upon Cul2_LRR1 inhibition in**
**TRAI^{-/-} cells. This seems little compared to WT cells. Here again, a clearer**
**way to represent quantifications would help compare data."**

The data in Figure EV4F illustrate that TRAI^{-/-} cells grow more slowly than
wild type cells (doubling time of ~17 hours compared to 12.5 hours). This likely

explains the slightly lower proportion of cells that accumulate GFP-SLD5 on
heterochromatin patches, following transient inhibition of CUL2-LRR1 (discussed
now on lines 376-378).

REVIEWER 2

The reviewer summarised her / his view as follows: *“the authors present the*
*first demonstration on the existence of two regulatory pathways and on the*
*role of p97 ATPase in the disassembly of the mammalian replisome. In*
*addition, they reveal mouse ES cells as tractable model system for studying*
*the disassembly of mammalian replisome. And the data in this study are very*
*clear and the manuscript is well-organized and well-written to easily follow.”*
*“To my point of view, the introduction of mouse ES cells as a tractable system*
*accompanied by gene editing method in order to study mammalian replisomes*
*in spontaneous and drug-treated conditions will be the major contribution*
*since it can be a valuable tool kit to the future research in studying replisomes*
*in mammalian cells. I believe this manuscript be publishable in your journal*
*after some minor revisions.”*

*“These findings will contribute to significant advances on our understanding*
*of mammalian DNA replication processes, including CMG disassembly.”*

Minor points to address:

1. *“Presenting a descriptive model on CMG disassembly in mammalian cells*
*will be helpful for general readers in following the manuscript.”*

Models for CMG disassembly in mammalian cells during DNA replication
termination (A) and during mitosis (B-C) are now shown in Figure EV5.

2. *“Even though statistics are depicted in the method section, it would be*
*better to also describe statistics in more details (p values, SD, S.E.M. and so*
*on) in figure legends if applicable.”*

Details of statistics have now been included where applicable in the figure
legends of the revised manuscript (legend to Figure 5D, previously Figure 4D and
legend to Figure EV4, previously Figure S5).

3. *“In figure 2C, please add (-) in the last panel on the top.”*

Thank you – we have now corrected this error in Figure 3C (formerly Figure 2C).

4. *“Is it possible to show colored images instead of black/white images? If*
*possible, it should be better to show colored images.”*

It is generally accepted that single-channel images should be presented in
grayscale (e.g. Johnson, J., *Mol. Biol. Cell*, 2012, 23, 754-757). This is the
convention that we have followed. In addition, however, we have now included a
pseudocolour merge for Figure 2C.

For greater clarity and to provide visual guides to the reader, we have also
changed the colour of all the labels in each panel of the Figures containing
microscopy data, so that the text ‘GFP-SLD5’ is coloured green, mCherry-PCNA is
coloured red, etc.

5. *“Please, describe how to generate stable cells expressing mCherry-H2B and*
*mCherry-PCNA in mES cells in the method section.”*

mCherry-H2B was expressed from the CAG promoter at the ROSA26 locus.
Details are now provided in the legend to Figure 4B (lines 2783-2784) and in
Materials and Methods (lines 542-550).

mCherry-PCNA was expressed from the CMV promoter, via random
integration of a linearised plasmid (pcDNA3.1-mCherry-PCNA, see Appendix Table
S1) that also contained a kanamycin / G418 resistance gene. Transfected cells were
selected for 9 days with medium containing 300 µg / ml G418, before single-cell
sorting via flow cytometry. Stable clones were then monitored by immunoblotting
and spinning disk confocal microscopy. Details are provided in Materials and
Methods (lines 561-568).

REVIEWER 3

**The reviewer summarised her / his view as follows: “The cellular system**
**described in this study is well designed and yields clear results: using**
**CRISPR/Cas9, the authors have introduced TAP or GFP tags in SLD5, a**
**component of CMG, without disrupting the endogenous regulation of SLD5**
**gene expression. Then, chromatin association/ and dissociation of the CMG**
**complex is monitored through confocal microscopy analysis of SLD5 at dense,**
**heterochromatic DNA regions that are replicated in late S phase. The**
**biochemistry experiments showing the integrity of the CMG complex and/or**
**MCM7 ubiquitylation in different experimental conditions (e.g. Fig. 1A, Fig. 2A-**
**C, Figs. S2 and S3) are flawless.**

**In my opinion, the results strongly support the conclusions of the paper. While**
**a role of CUL2-LRR1 and TRAIIP ubiquitylation pathways in CMG disassembly**
**could possibly be anticipated from the previous work in model systems, it is**
**nicely demonstrated here with convincing data.”**

**“I do not have any major criticism about experimental design or execution.”**

**“The methods developed here with mESCs also open the road for future**
**analyses of the different proteins involved in this process.**

**The discussion could benefit from some additional speculation about the**
**relative strength/reliability of the CUL2-LRR1 vs TRAIIP pathways, why this**
**dual system is evolutionary conserved, etc.”**

We have added an additional figure to the revised manuscript (Figure EV5)
with models to illustrate the proposed roles of CUL2^{LRR1} and TRAIIP in the
mammalian cell cycle. These are discussed between lines 410-418, making clear
that CUL2^{LRR1} mediates CMG disassembly during DNA replication termination,
whereas TRAIIP-dependent CMG disassembly during mitosis is likely to have
evolved to allow metazoan cells with large genomes to process unreplicated DNA
during mitosis, in order to allow the completion of nuclear division.

**“My only other comment is that a recent study from the Walter laboratory (Wu**
**et al, 2019, Nature) has reported the role of Xenopus TRAIIP and p97 in CMG**
**ubiquitylation and eviction in the context of fork convergence at DNA inter-**
**strand crosslinks. This biological context shares many similarities with**
**replication termination events, and I feel that this study should be referenced**
**and discussed.”**

We now include discussion of this point on lines 441-444 and cite the paper
by Wu et al.

Dear Karim,

Thank you for the transfer of your revised manuscript. We have now received the enclosed report from referee 1, who was asked to assess it. This referee still has minor suggestions that I would like you to incorporate before we can proceed with the official acceptance of your manuscript.

A few editorial changes are also required:

- The reference format lists more than 10 authors, please correct. A maximum of 10 authors before "et al" should be listed.
- Please send us a completed author checklist that can be found here: <https://www.embopress.org/page/journal/14693178/authorguide>. The checklist will also be part of the transparent peer-review process file (RPF).
- Please upload all figures as individual files.
- Fig 2A is called out before Fig 1H. Fig 5A is called out after 5D. Appendix Figs S1+S2 panels are not called out. Please correct.
- Please upload the Appendix as a separate pdf file with a table of content and page numbers. The Appendix table needs to be called Appendix Table S1.
- Fig 1J seems to contain a splice. Please send us the source data for this figure panel.

I attach to this email a related manuscript file with comments by our data editors. Please address all comments in the final manuscript.

EMBO press papers are accompanied online by A) a short (1-2 sentences) summary of the findings and their significance, B) 2-3 bullet points highlighting key results and C) a synopsis image that is exactly 550 pixels wide and 200-600 pixels high (the height is variable). You can either show a model or key data in the synopsis image. Please note that text needs to be readable at the final size. Please send us this information along with the revised manuscript.

Referee #1:

The revision of the manuscript "CUL2LRR1, TRAIIP and p97 control CMG helicase disassembly in

the mammalian cell cycle" largely erased the few concerns raised regarding the original manuscript version. The data is clear and convincing and supports the conclusions presented. I fully support publication of the manuscript.

Two minor points could be addressed:

1) Quantification of microscopy images (typically % cells) are currently presented in the main text in brackets. This makes it difficult to compare numbers within an experiment and between experiments, as I stated in my first review. The authors should indicate the numbers in the figures, either as graphs or numbers in the images shown.

2) In line 236, the conclusion is presented that the observation that CMG remains on chromatin in the presence of p97i indicates that p97 is required for CMG disassembly during termination. Although this is likely, an alternative explanation is that replisomes stall upon p97 inhibition and the termination stage is never reached. If the authors do not disagree with this possibility they should mention this alternative explanation, perhaps also saying why it is less likely. I apologise that I did not raise this point in my first review.

**RESPONSE TO REFEREE COMMENTS:**

The referee raised two final points:

1. ***“Quantification of microscopy images (typically % cells) are currently***
***presented in the main text in brackets. This makes it difficult to compare***
***numbers within an experiment and between experiments, as I stated in my first***
***review. The authors should indicate the numbers in the figures, either as***
***graphs or numbers in the images shown.”***

Quantification has now largely been moved from main text to the Figures and Figure
legends (in many cases the latter makes sense for clarity and due to space
limitations in the Figure panels).

2. ***“In line 236, the conclusion is presented that the observation that CMG***
***remains on chromatin in the presence of p97i indicates that p97 is required for***
***CMG disassembly during termination. Although this is likely, an alternative***
***explanation is that replisomes stall upon p97 inhibition and the termination***
***stage is never reached. If the authors do not disagree with this possibility they***
***should mention this alternative explanation, perhaps also saying why it is less***
***likely. I apologise that I did not raise this point in my first review.”***

The data argue strongly that p97 inhibition leads to persistence of ubiquitylated CMG
helicase on chromatin, reflecting the role of p97 in disassembly of ubiquitylated
CMG. We do not agree that the data are consistent with replisome stalling, for two
main reasons.

- firstly, Figure 2A shows that p97 inhibition leads to the accumulation of CMG with
ubiquitylated MCM7 subunit. Ubiquitylation occurs specifically during DNA
replication termination, not during replisome stalling.

- secondly, time-lapse data in Figure EV3 shows that mCherry-PCNA still associates
transiently with heterochromatic patches in late S-phase upon inhibition of p97,
indicating that replication kinetics are indistinguishable from the control (no evidence
of stalling). In contrast to PCNA (marker of ongoing DNA synthesis), GFP-SLD5 and
other core-replisome factors remain on chromatin subsequently (indicating
persistence of the replisome complex, but not stalling of DNA synthesis).

Considering together the above two points, the most reasonable conclusion is that
p97 inhibition leads to the persistence of ubiquitylated CMG helicase on chromatin
after DNA replication termination. Therefore, we have not adjusted the relevant
section of the text (lines 186-192 and 225-232).

Karim Labib
University of Dundee
MRC Protein Phosphorylation and Ubiquitylation Unit
Sir James Black Centre
Dow Street
Dundee DD1 5EH
United Kingdom

Dear Karim,

I am very pleased to accept your manuscript for publication in the next available issue of EMBO reports. Thank you for your contribution to our journal.

At the end of this email I include important information about how to proceed. Please ensure that you take the time to read the information and complete and return the necessary forms to allow us to publish your manuscript as quickly as possible.

As part of the EMBO publication's Transparent Editorial Process, EMBO reports publishes online a Review Process File to accompany accepted manuscripts. As you are aware, this File will be published in conjunction with your paper and will include the referee reports, your point-by-point response and all pertinent correspondence relating to the manuscript.

If you do NOT want this File to be published, please inform the editorial office within 2 days, if you have not done so already, otherwise the File will be published by default [contact: emboreports@embo.org]. If you do opt out, the Review Process File link will point to the following statement: "No Review Process File is available with this article, as the authors have chosen not to make the review process public in this case."

Should you be planning a Press Release on your article, please get in contact with emboreports@wiley.com as early as possible, in order to coordinate publication and release dates.

Thank you again for your contribution to EMBO reports and congratulations on a successful publication. Please consider us again in the future for your most exciting work.

Best wishes,
Esther

THINGS TO DO NOW:

You will receive proofs by e-mail approximately 2-3 weeks after all relevant files have been sent to our Production Office; you should return your corrections within 2 days of receiving the proofs.

Please inform us if there is likely to be any difficulty in reaching you at the above address at that time. Failure to meet our deadlines may result in a delay of publication, or publication without your corrections.

All further communications concerning your paper should quote reference number EMBOR-2020-52164V2 and be addressed to emboreports@wiley.com.

Should you be planning a Press Release on your article, please get in contact with emboreports@wiley.com as early as possible, in order to coordinate publication and release dates.

Corresponding Author Name: KARIM LABIB

Journal Submitted to: EMBO REPORTS

Manuscript Number: EMBOR-2020-52164V1